# The Small Molecule GAL-201 Efficiently Detoxifies Soluble Amyloid β Oligomers: New Approach towards Oral Disease-Modifying Treatment of Alzheimer’s Disease

**DOI:** 10.3390/ijms23105794

**Published:** 2022-05-21

**Authors:** Hermann Russ, Michele Mazzanti, Chris Parsons, Katrin Riemann, Alexander Gebauer, Gerhard Rammes

**Affiliations:** 1Galimedix Therapeutics Inc., 2704 Calvend Lane, Kensington, MD 20895, USA; hruss@galimedix.cm (C.P.); agebauer@galimedix.com (A.G.); 2Laboratory of Cellular and Molecular Physiology, Department of Biosciences, University of Milano, Via Celoria 26, 20133 Milano, Italy; michele.mazzanti@unimi.it; 3Department of Anaesthesiology and Intensive Care Medicine, Technical University Munich, Ismaningerstr. 22, 81675 Munich, Germany; katrin.riemann@mri.tum.de (K.R.); g.rammes@tum.de (G.R.)

**Keywords:** LTP, synaptic plasticity, hippocampus, slice, beta amyloid, soluble oligomers, neurodegenerative disease, Alzheimer, GAL-201

## Abstract

Soluble amyloid β (Aβ) oligomers have been shown to be highly toxic to neurons and are considered to be a major cause of the neurodegeneration underlying Alzheimer’s disease (AD). That makes soluble Aβ oligomers a promising drug target. In addition to eliminating these toxic species from the patients’ brain with antibody-based drugs, a new class of drugs is emerging, namely Aβ aggregation inhibitors or modulators, which aim to stop the formation of toxic Aβ oligomers at the source. Here, pharmacological data of the novel Aβ aggregation modulator GAL-201 are presented. This small molecule (288.34 g/mol) exhibits high binding affinity to misfolded Aβ_1-42_ monomers (K_D_ = 2.5 ± 0.6 nM). Pharmacokinetic studies in rats using brain microdialysis are supportive of its oral bioavailability. The Aβ oligomer detoxifying potential of GAL-201 has been demonstrated by means of single cell recordings in isolated hippocampal neurons (perforated patch experiments) as well as in vitro and in vivo extracellular monitoring of long-term potentiation (LTP, in rat transverse hippocampal slices), a cellular correlate for synaptic plasticity. Upon preincubation, GAL-201 efficiently prevented the detrimental effect on LTP mediated by Aβ_1-42_ oligomers. Furthermore, the potential to completely reverse an already established neurotoxic process could also be demonstrated. Of particular note in this context is the self-propagating detoxification potential of GAL-201, leading to a neutralization of Aβ oligomer toxicity even if GAL-201 has been stepwise removed from the medium (serial dilution), likely due to prion-like conformational changes in Aβ_1-42_ monomer aggregates (trigger effect). The authors conclude that the data presented strongly support the further development of GAL-201 as a novel, orally available AD treatment with potentially superior clinical profile.

## 1. Introduction

Amyloid beta (Aβ) has been known for more than 100 years as a hallmark of the pathology underlying Alzheimer’s disease (AD). Various unsuccessful approaches with drug candidates have been taken in the past decades to remove Aβ from AD patients’ brain in order to treat the disease. However, recent results indicate that only very specific Aβ species are an appropriate drug target, namely soluble Aβ oligomers, which attack neurons in the brain and kill them, leading to progressive neurodegeneration, dementia, and finally to neuronal death [1,2].

Although the molecular mechanism by which Aβ exerts its cytotoxicity is still not completely clarified, it is widely accepted that the toxic species of Aβ is represented by its soluble oligomeric aggregates [2,3,4,5]. Neurotoxicity caused by oligomers of Aβ_1-42_, which is the most common form of Aβ, is believed to happen both at the membrane level [6,7] as well as inside the neuron after cytoplasmic inclusion [8,9,10]. Several membrane receptors have been identified to react to Aβ_1-42_ oligomers either increasing their activity or inhibiting their functions [11,12,13,14,15,16]. It is still controversial if Aβ_1-42_ oligomers can directly cause membrane instability and/or have the ability to form ion conducting channel in vivo. Certainly this is true for artificial lipid bilayer where Aβ_1-42_ is responsible for generating conductances directly proportional to the Aβ_1-42_ aggregation state [17].

The self-assembly process of Aβ is linked to a structural transition of a physiologically folded into a misfolded monomeric form with a predominant β-sheet secondary structure, which is prone to aggregate into oligomers. Only the small, soluble, and diffusible oligomers can disrupt synaptic plasticity, which is associated with binding to plasma membranes and changing excitatory–inhibitory balance, perturbing metabotropic glutamate receptors, prion protein, and other neuronal surface proteins, downregulating glutamate transporters, causing glutamate spillover, and activating extrasynaptic NMDA receptors [2,18]. Through this mechanism, Aβ oligomers—not monomers—affect long-term potentiation (LTP), a cellular correlate for synaptic plasticity that is thought to underlie memory, both in vitro and in vivo.

Based on the better understanding of the Aβ pathomechanism, Biogen’s AD drug Aduhelm^TM^ was developed and approved in June 2021 by the FDA for treatment of AD [19]. This drug is considered to target predominantly soluble Aβ_1-42_ oligomers that as a secondary effect leads to a reduction in high-molecular Aβ deposits in the brain. The clinical data with Aduhelm^TM^ are not fully convincing: the drug shows a limited clinical effect size and is poorly tolerated with around one-third of the patients experiencing antibody-typical side effects (e.g., ARIA) [20,21]. Nevertheless, Aβ oligomers can now be considered as the first FDA-approved drug target for disease-modifying treatment of AD.

Several reports have shown that Aβ_1-42_ deposits are also present in different retina cell layers and in the optic nerve [22,23]. For example, it is not unusual that Alzheimer’s patients also suffer from degeneration of the retina [24,25]. Glaucoma and age-related macular degeneration (AMD) are the most recognized causes of irreversible vision loss worldwide. Indeed, both these pathological conditions show Aβ_1-42_ deposits in the retina [26,27].

A detailed understanding of the formation of toxic Aβ oligomers and their effects is essential to develop urgently required new treatments for AD patients. This manuscript describes pharmacological characteristics of the new drug candidate GAL-201 as an Aβ aggregation modulator based on data from binding experiments, from a rat pharmacokinetic study and from patch clamp and LTP experiments. We present data on the detoxifying effects of GAL-201 on oligomers formed from Aβ_1-42_, and on pyroglutamate-modified Aβ (AβpE3) [28,29,30,31,32]. AβpE3 was selected since a recent Phase 2 clinical AD study showed a significant positive outcome for the antibody donanemab, which targets AβpE3 [33].

## 2. Results

The target of GAL-201 is the misfolded monomeric β-sheet form of Aβ. The affinity of GAL-201 to immobilized Aβ_1-42_ was determined using in vitro SPR technology. The K_D_ is the measure for the target affinity and expresses the concentration of GAL-201 that elicited one-half of the maximum binding. The K_D_ of GAL-201 for Aβ monomers is 2.5 ± 0.6 nM (n = 4) (Figure 1).

### 2.1. Pharmacokinetics

GAL-201 is well absorbed in rats after oral administration and enters the brain interstitial fluid (ISF) in pharmacologically active concentrations (Figure 2A). Only 10 min after oral application of 50 mg/kg GAL-201, the plasma concentration was approximately 5 µM. The plasma level reached a maximum (C_max_) of 7.4 µM ± 0.9 (n = 4) at 150 min. The ISF concentration as measured by microdialysis increased gradually up to 505 ± 124 nM at 180 min. Comparing the concentrations of GAL-201 in the plasma and in ISF results in a factor of 15, meaning that around 7% of the systemically available compound crosses the blood–brain barrier after single dose application based on C_max_.

In addition, the time courses of the GAL-201 plasma concentration after oral administration of 25 mg/kg and 10 mg/kg have been determined (data not presented here). Analysis of these AUC_0–24h_ values and the C_max_ values were compatible with dose proportionality. Model calculations revealed an oral bioavailability of 16–24% in rats.

Since 50 mg/kg GAL-201 oral dose required an intestinal resorption time of more than 2 h the elimination kinetics were difficult to analyze. Therefore, additional studies have been performed to investigate the pharmacokinetics of smaller doses injected subcutaneously (s.c.). Figure 2B shows the time course of the ISF concentration of GAL-201 as measured by microdialysis after s.c. injection of 0.4 mg/kg and 2.0 mg/kg. GAL-201 (2.0 mg/kg) showed a peak concentration 40 min after injection of 60 nM regardless of whether the animals were anesthetized or were able to move freely. The low dose of 0.4 mg/kg revealed peak concentration of 10 nM GAL-201 in ISF, which is consistent with dose proportionality. The elimination half-life of GAL-201 from ISF is in the range of 2 h. Since GAL-201 (10 nM) has been shown in the LTP experiments to be efficacious, the oral target dose in rats is defined as 1 mg/kg.

GAL-201 was also screened at 100 µM against 61 anti-targets (Ricerca Panel: enzymes, kinases, transporters, hormones, factors, and chemokines) without any hits.

### 2.2. Measuring Isolated Neurons Resting Potential

The impact of Aβ_1-42_ on an isolated neuron was tested by monitoring the cell resting membrane potential using electrophysiology techniques. Whole cell, perforated patch experiments were carried out on isolated neonatal mice hippocampal neurons. Aβ_1-42_ was acutely perfused on a single cell under continuous membrane potential recording. Figure 3A shows two examples of time courses of cell membrane voltage during perfusion (arrows) of 50 nM Aβ_1-42_ (left) and the same concentration of Aβ_1-42_ but in the presence of 1 µM of GAL-201 (right). In panel B, averages of resting membrane potential recordings in the different conditions are shown. GAL-201 at a concentration of 1 µM (only tested dose) prevents neuronal cell depolarization, maintaining almost constant the resting membrane potential at the original level.

### 2.3. In Vitro LTP

GAL-201 was used at various concentrations in hippocampal CA1-LTP. Surprisingly, when GAL-201 was applied at the lowest concentration of only 10 nM, it prevented the detrimental effect of 50 nM Aβ_1-42_ on hippocampal CA1-LTP (LTP with GAL-201 = 1.42 ± 0.17, n = 7; 50 nM Aβ_1-42_ = 1.10 ± 0.13, n = 12 vs. Aβ_1-42_/GAL-201 = 1.33 ± 0.11, n = 7; *p* = 0.0001, 95% CI = 0.16 to 0.49; Figure 4A,B). Figure 4C summarizes the effects of different stoichiometric Aβ _1-42_/GAL-201 ratios (10:1, 2:1, 1:5) on their ability to prevent LTP reduction. HFS elicited an fEPSP potentiation of 1.51 ± 0.18, (n = 10) and Aβ _1-42_ reduced LTP to 1.10 ± 0.10 (n = 13), *p* < 0.0001, 95% CI = 0.30 to 0.60. In comparison to the effect with Aβ _1-42_ alone, GAL-201 (500 and 10 nM) significantly prevented LTP (1.24 ± 0.13, n = 10, *p* = 0.014, 95% CI = −0.33 to −0.029 and 1.33 ± 0.11, n = 7, *p* = 0.0004, −0.44 to −0.11, respectively), whereas GAL-201 (100 nM) in the presence of Aβ _1-42_ (50 nM) elevated fEPSP potentiation, but not significantly (*p* = 0.12, n = 6, 95% CI = −0.59 to −0.13).

The results of the experiments presented in Figure 4 and other experiments with higher GAL-201 concentrations are summarized in Table 1. The control is defined as 100% normalized LTP response and the normalized LTP change under the toxic effect of 50 nM Aβ_1-42_ oligomers is defined as 0%. Using the usual stoichiometric excess for GAL-201, namely 500 or 100 nM, a moderate detoxifying effect of 40.2% and 34.7% was observed, respectively. However, reversing the stoichiometric ratio and using GAL-201 at a concentration of only 10 nM, representing a five-fold stoichiometric undersupply, significantly improves the detoxifying strength, reaching 60% effect size.

The principle of “serial dilution” is explained in Figure 5. The final solution obtained by serial dilution, which contained only traces of GAL-201 (0.1 nM) and 50 nM of detoxified Aβ_1-42_ (in analogy AβpE3) in tube #5, is indicated here as GAL-201/Aβ_1-42_-SD5 (or GAL-201/AβpE3-SD5). In Figure 6, we demonstrate that GAL-201/Aβ_1-42_SD5 clearly prevented the synaptotoxic effect of Aβ_1-42_ on the hippocampal CA1-LTP (LTP control = 1.86 ± 0. 12, n = 10; with 50 nM Aβ_1-42_ = 1.09 ± 0.06, n = 6; in the presence of Aβ_1-42_/0.1 nM GAL-201 = 1.68 ± 0.12, n = 10; Aβ_1-42_ vs. Aβ_1-42_/GAL-201: *p* = 0.0077, 95% CI = −0.04 to −0.15; Figure 6). Interestingly, this post-serial dilution solution was also effective against the detrimental effect of AβpE3 (LTP control = 1.37 ± 0.21, n = 3; with 50 nM AβpE3 = 0.98 ± 0.02, n = 3; 0.1 nM GAL-201 = 1.41 ± 0.04, n = 9; in the presence of Aβp E3/GAL-201 = 1.34 ± 0.05, n = 9; AβpE3 vs. GAL-201/AβpE_3_SD5: *p* = 0.0017, 95% CI = −0.04 to −0.15; Figure 7).

In the experimental setting described above, either the GAL-201/Aβ_1-42_-SD5 or GAL-201/AβpE3 SD5 solution was applied after establishing a control LTP in the presence of 0.1 nM GAL-201, which resembles a prophylactic or preventive treatment situation. Mimicking the clinical situation needs an already initiated neurotoxic process in that receptor modulation and activation of downstream cascades by Aβ were already set up by the time of starting the application with GAL-201 and can then be detoxified. Therefore, to elucidate whether GAL-201, when serial diluted with Aβ_1-42_ is capable of detoxifying an already established neurotoxic process, the first LTP was induced in the presence of 50 nM Aβ_1-42_ and the second after the application of GAL201/Aβ_1-42_SD5. Figure 8 shows the pooled data of the full-length two-input experiments for GAL-201. After slices were incubated with GAL201/Aβ_1-42_SD5, LTP completely recovered to 1.99 ± 0.42 (n = 9). LTP was reduced by Aβ_1-42_ alone to 1.14 ± 0.15 (n = 10). This effect is shown also as a scatter dot plot in Figure 8B (Aβ_1-42_ vs. GAL201/Aβ_1-42_SD5 *p* < 0.0001 *t*-test). This detoxifying effect is not simply the result of a decrease in Aβ_1-42_ neurotoxicity with time, as shown in our recent publication [34]. Here, LTP was blocked by Aβ_1-42_ to 1.13 ± 0.6 (n = 10) and, after removal of Aβ_1-42_ for 90 min, HFS did not produce LTP either (1.09 ± 0.5, n = 10; data not shown). These experiments clearly show that even after removal of Aβ_1-42_ for 90 min, the Aβ_1-42_-induced neurotoxic processes were still ongoing and did not allow an EPSP potentiation after HFS.

### 2.4. In Vivo LTP

Preliminary studies tested the effects of a 10 mg/kg and a 2 mg/kg dose of s.c. administered GAL-201 on TBS-induced hippocampal LTP, in the absence of oligomeric Aβ_1-42_. GAL-201 10 mg/kg itself conferred an inhibitory effect on hippocampal synaptic responses following theta burst stimulation: GAL-201 10 mg/kg GAL-201 s.c. treatment = 129.2 ± 3.5% of baseline (n = 3) 80–90 min following LTP induction, compared to 179.2 ± 8.7% in s.c. vehicle-treated animals (n = 12, *p* < 0.05, data not shown). No further studies with the 10 mg/kg dose were conducted. GAL-201 2 mg/kg itself did not confer any significant inhibitory or facilitatory effect on hippocampal synaptic responses following theta burst stimulation: GAL-201 2 mg/kg s.c. treatment = 169.5 ± 7.0% of baseline (n = 7) 80–90 min following LTP induction, compared to 179.2 ± 8.7% in s.c. vehicle-treated animals (n = 12, Figure 9A,C). Further studies with GAL-201 were conducted with a maximal dose of 2 mg/kg.

In animals subject to *i.c.v* injection of 6 µL oligomeric Aβ_1-42_ preceded by s.c. vehicle, significant deficits in LTP were observed. At the end of the recording period, 80–90 min after LTP induction, the PS amplitude following i.c.v. administration of oligomeric Aβ_1-42_ measured 138.1 ± 4.1% of baseline (n = 18), compared to 179.2 ± 8.7% in PBS-injected animals (n = 12, Figure 9B,C and Figure 10).

In animals subject to *i.c.v* injection of 6 µL oligomeric Aβ_1-42_, a pre-administration of s.c. GAL-201 0.08–2 mg/kg produced a dose-dependent protection against oligomeric Aβ_1-42_-induced deficits in hippocampal LTP, the highest GAL-201 doses (0.4 mg/kg and 2 mg/kg) bringing post-TBS responses towards those observed in animals receiving i.c.v. PBS (*p* < 0.05, one-way ANOVA with post hoc Bonferroni test; Figure 9A,B and Figure 10). At the end of the recording period, 80–90 min after LTP induction, the PS amplitude following s.c. administration of GAL-201 and i.c.v. administration of oligomeric Aβ_1-42_ measured 170.0 ± 11.9% of baseline (n = 7) for the 2 mg/kg dose (Figure 9C and Figure 10), 174.7 ± 16.0% of baseline (n = 6) for the 0.4 mg/kg dose (Figure 9C and Figure 10), and 139.6 ± 13.1% of baseline (n = 4) for the 0.08 mg/kg dose (Figure 10), compared to 179.2 ± 8.7% in vehicle + PBS-injected animals (n = 12) and 138.1 ± 4.1% in vehicle and i.c.v. oligomeric Aβ_1-42_–injected animals (n = 18, Figure 9 and Figure 10).

Table 1 shows that the reversal of the stoichiometric ratio between GAL-201 and Aβ_1-42_ increases the detoxifying strength GAL-201. The data in this table are taken from the experiment presented in Figure 4. The control experiment is defined as 100% normalized LTP response and under the toxic effect of 50 nM Aβ_1-42_ oligomers, the normalized LTP change is defined as 0%. Using the usual stoichiometric excess for GAL-201, namely 500 or 100 nM, a moderate detoxifying effect of 40.2% and 34.7% was observed, respectively. However, reversing the stoichiometric ratio and using GAL-201 at a concentration of only 10 nM, representing a five-fold stoichiometric undersupply, significantly improves the detoxifying strength reaching 60%.

## 3. Discussion

Alzheimer’s disease (AD) is one of the largest global health challenges with huge implications for individuals and society [35]. AD prevalence is growing at alarming rates worldwide and is associated with the aging of the population. Already more than 50 million people worldwide suffer from AD, with that number expected to rise to around 150 million by 2050 if no effective disease-modifying treatment can be introduced [36]. The currently available treatment options are not satisfactory. The recently launched first disease-modifying AD drug Aduhelm^TM^ will not improve the situation significantly because the drug has only a weak treatment effect, poor tolerability, is administered by injection, and is not affordable for most patients worldwide [20,21]. Nevertheless, owing to aducanumab, Aβ oligomers can now be considered as the first FDA-approved drug target for disease-modifying treatment of AD.

New treatment approaches are urgently needed to fight AD. A promising new AD drug could target Aβ oligomers; however, significantly improved efficacy is needed. In addition, this new drug should be well tolerated, easy to be self-administered (such as in a tablet), and easy to manufacture in order to permit a reasonable price for global markets. Here, we describe the pharmacological profile of a new small molecule, GAL-201, that has the potential to meet these criteria.

### 3.1. Targeting Aβ_1-42_ Oligomers

Regular Aβ monomers are needed for regular synaptic function and should not be eliminated from brain tissue [37,38]. When Aβ monomers are overexpressed in a pathological condition, they tend to misfold exposing the VFFAEDVGSNK motif (amino acids 18–28) within the Aβ sequence. Misfolded Aβ monomers then start to aggregate, first to soluble toxic Aβ oligomers, later to aggregates with higher molecular weight and β-sheet conformation that precipitate in the tissue and finally appear as Aβ deposits and plaques. By analogy to studies based on coupling energy values using the structurally closely related molecule GAL-101 (formerly MRZ-99030) [39], GAL-201 is thought to bind preferentially to the misfolded form of Aβ_1-42_. The SPR binding data presented here show a high-affinity binding of GAL-201 with a K_D_ of 2.5 ± 0.6 nM (n = 4), which supports the further development of this small molecule as a drug. The high binding affinity of GAL-201 to Aβ_1-42_ monomers appears to be unmatched by any other small molecule Aβ aggregation modulator.

### 3.2. Availability of GAL-201 to Enter the Brain after Oral Application

Pharmacokinetic studies using plasma and brain dialysate were performed in rats to determine whether the small molecule GAL-201 has potential for clinical development as an oral drug. GAL-201 is sufficiently absorbed in the intestine of rats to reach pharmacologically active concentrations in the brain (Figure 2A). Model calculations reveal that an oral dose in rats of approximately 1 mg/kg is sufficient to achieve a brain exposure (10 nM peak concentration) that triggers the self-propagating effect of GAL-201 in eliminating toxic Aβ oligomers. It is not possible to predict the oral doses for humans from animal studies; however, the bioavailability in humans is typically even better due to the larger intestinal surface of humans relative to rats. Overall, the available PK data in rats strongly support initiation of a human Phase 1 study with oral GAL-201 administration after completion of the IND-enabling program.

### 3.3. Patch Clamp with Hippocampal Neurons

Especially at early stages of AD, the hippocampus, a brain area critical for learning and memory, is vulnerable to neurodegenerative processes associated with Aβ [40]. Therefore, studying the effects of toxic Aβ_1-42_ oligomers in hippocampal neurons is instructive to better understand the mechanism of GAL-201 action. Patch clamp experiments were carried out on isolated neonatal mice hippocampal neurons. Toxic Aβ_1-42_ oligomers generated from 50 nM Aβ_1-42_ depolarize the cells from −70 to −25 mV. With such a low potential, neurons can no longer operate physiologically and, at some point, become apoptotic. The mechanism of a direct cytotoxic effect of Aβ on isolated neurons is not yet fully elucidated. Several studies have investigated how the binding of the Aβ oligomers to NMDA receptors leads to Ca^2+^ channel opening [16,41,42]. In addition, a direct interaction between the neuronal plasma membrane and Aβ oligomer species has been described, leading to the formation of ion-permeable pores in the lipid bilayer [43]. Both processes could cause drastic membrane depolarization with deadly long-term consequence for these neuronal cells.

GAL-201 has been applied in these experiments at a probably supra-effective dose of 1 µM, since at that time the true potency of GAL-201 had still been underestimated. GAL-201 at that concentration had no effect by itself in the patch clamp setting; however, it almost completely suppressed the toxic effect of the Aβ_1-42_ oligomers on membrane potential (Figure 3). In addition to the LTP experiments, these results provide additional independent in vitro evidence supportive of the Aβ_1-42_ oligomers detoxifying effect of GAL-201, even if the concentrations actually required to induce that effect (potency) are unknown.

### 3.4. In Vitro LTP Using Aβ_1-42_ or AβpE3

Long-term potentiation (LTP) is regarded as an electrophysiological correlate of learning and memory and is impaired by acute administration of Aβ_1-42_ oligomers, e.g., to hippocampal slices [44,45]. In vitro LTP experiments were carried out using 50 nM Aβ_1-42_ oligomers to suppress LTP response and various concentrations of GAL-201 to antagonize that toxic effect. GAL-201 at 500 nM reduced, but did not block LTP induction per se, probably due to an excessive dose compared to its potency. This LTP-suppressive property of 500 nM probably explains why this concentration of GAL-201 showed only moderate effects on LTP (Figure 5C). Importantly, after reducing the GAL-201 concentration to 100 nM, LTP was no longer affected; however, 100 nM GAL-201 was not able to prevent the Aβ_1-42_-induced synaptotoxicity on LTP significantly. Interestingly, at a concentration of only 10 nM, the toxic effect of Aβ_1-42_ was prevented more strongly and reached high significance. This atypical dose response is summarized in Table 1.

For protein–protein interactions, the stoichiometric ratio between both components is important. This also applies to the interaction between an Aβ aggregation modulator such as GAL-201 and its target Aβ_1-42_. Using the usual stoichiometric excess of GAL-201 over the 50 nM Aβ_1-42_, namely 500 or 100 nM, a moderate detoxifying effect of 40.2% and 34.7% (not significant) was observed, respectively (Table 1). However, reversing the stoichiometric ratio and using GAL-201 at a concentration of only 10 nM, representing a five-fold stoichiometric undersupply, significantly improves the detoxifying strength reaching 60.0%. Future experiments should clarify at which stoichiometric ratio the detoxifying effect is most pronounced. Nevertheless, this finding contrasts with the previously prevailing assumption that multiple molecules of the aggregation modulator are required to detoxify one amyloid beta molecule [46,47]. For the compound GAL-101, which is a structurally related Aβ aggregation modulator, at least ten molecules are needed for the inhibition of one Aβ_1-42_ molecule (stoichiometric ratio 10:1). For other Aβ aggregation modulators such as ALZ-801 and PRI-002 [47,48], several hundred or thousand molecules are needed for inhibiting the toxicity caused by one Aβ_1-42_ molecule (stoichiometric ratio in the range of approximately 100:1 to 5000:1). In contrast, one molecule GAL-201 was demonstrated to antagonize approximately five Aβ_1-42_ molecules. This efficiency is unprecedented and we expect that it would translate in AD patients into a strong Aβ_1-42_ detoxifying effect at rather low oral doses.

### 3.5. Self-Propagating Detoxification of Aβ_1-42_ Using Serial Dilation of GAL-201

The principle of serial dilution is illustrated in Figure 3 and has been published before [34]. At the end of the dilution series, vial #5 contains a pharmacologically inactive trace amount of 0.1 nM GAL-201 in the presence of 50 nM Aβ_1-42_. For this solution, the term “GAL-201/Aβ_1-42_ SD5” was introduced. Using GAL-201/Aβ_1-42_ SD5 solution in the in vitro LTP experiments (Figure 6) leads to an almost complete detoxification of the soluble Aβ_1-42_ oligomers. Since this cannot be due to the trace amount of GAL-201 (0.1 nM GAL-201 have been demonstrated to be inactive without serial dilution), we hypothesize that it is due to a self-propagating, self-detoxifying conformation of Aβ_1-42_ that has been formed at the start of the cascade in presence of GAL-201 and propagated itself again at each dilution step. This prionlike phenomenon has been described as a “seeding effect” and “trigger effect” for MRZ-99030 [34,49]. Here, we demonstrate that this trigger effect is also induced by GAL-201.

GAL-201/Aβ_1-42_ SD5 could also detoxify the Aβ isoform AβpE3 (Figure 7), which has also been shown to exert synaptotoxic effects against LTP [45]. AβpE3 is a clinically relevant target for AD-like Aβ_1-42_ since the anti-AβpE3 antibody donanemab demonstrated clinical efficacy in AD patients in a recent Phase II study [33]. Thus, we expect that GAL-201 has the potential to detoxify both Aβ_1-42_ and AβpE3 in the brain of AD patients, combining the beneficial effects of the antibodies aducanumab and donanemab.

### 3.6. Reversal of an Already Established Synaptotoxic Effect Caused by Aβ_1-42_

In additional in vitro LTP experiments (Figure 8), brain slices were first exposed to toxic oligomers generated from 50 nM Aβ_1-42_, resulting in a complete loss of the LTP response. Replacement of the medium with GAL-201/Aβ_1-42_SD5 solution completely reversed the LTP signal to normal. This experiment mimics the situation in AD patients in whom the toxicity of Aβ_1-42_ had been affecting the neurons before treatment initiation. Immediate cessation of pre-existing synaptotoxicity by GAL-201 treatment can be expected to rapidly restore synaptic plasticity and thereby improve neuronal function, especially cognition.

### 3.7. In Vivo Effects of GAL-201 Using LTP

Compared to the in vitro *LTP* described above, the in vivo experiments are one step closer to the situation in AD patients since GAL-201 is administered systemically (here s.c.). The soluble Aβ_1-42_ oligomers were injected directly into the brain ventricles (i.c.v.) of the rats at concentrations inducing pronounced LTP deficits (Figure 10). The major finding was that the s.c. dose of 0.4 mg/kg GAL-201 resulting in a GAL-201 peak concentration of 10 nM in the extracellular space of the brain (Figure 2B) showed a complete detoxification of the i.c.v. injected Aβ_1-42_ (normalized LTP). This peak concentration was reached at the time point of the i.c.v. injection of Aβ_1-42_. Those data are consistent with the in vitro LTP results described above, where 10 nM GAL-201 also showed significant prevention of Aβ_1-42_ synaptotoxicity. Thus, the targeted peak concentration of GAL-201 in the extracellular space of AD patients’ brains should be in the range of 10 nM. This concentration is capable of detoxifying the Aβ_1-42_ (probably also the AβpE_3_) oligomers and induces the trigger effect for a long-lasting self-propagating transformation of toxic Aβ_1-42_ oligomers into nontoxic Aβ species. In the forthcoming Phase 1 study of oral GAL-201 administration, CSF sampling will ensure a concentration of 10 nM GAL-201 is reached but not exceeded in the extracellular space of the brain.

### 3.8. Comparison of GAL-201 with Other Aβ Oligomer Targeting Drugs

There are currently several Aβ aggregation modulators under development. Most advanced is ALZ-801, which is currently in clinical Phase 3 focusing on ApoE4/4 AD patients. ALZ-801 is a pro-drug of tramiprosate, an endogenous small molecule from the human brain that inhibits the aggregation of Aβ_1-42_ into oligomers [50]. Compared to GAL-201, the affinity of ALZ-801 to Aβ_1-42_ is rather low and the important self-propagating effect of Aβ detoxification has not been described for it. PRI-002 is a peptide with 12 D-amino acid residues. It has recently completed Phase 1 and has a similar mode of action to GAL-201 in fighting toxic Aβ oligomers, including the self-propagating detoxification mechanism [51]. The molecule can be administered orally, however, with a low bioavailability typical of peptide drugs. Compared to GAL-201, the affinity of PRI-002 for Aβ_1-42_ is lower by a factor of around 1000. GAL-101 has a similar mechanism of action to GAL-201 [39]. It is under development for ophthalmology indications, specifically for glaucoma and dry AMD, two diseases associated with as Aβ_1-42_-induced neurodegeneration in the retina [27]. GAL-101 has completed a Phase 1 study with eye-drop administration. This compound has also been shown to seed a self-replication of nontoxic Aβ aggregates [34].

Compared with the Aβ oligomer-targeting antibody aducanumab (FDA-approved as Aduhelm^TM^), the small molecule GAL-201 can enter the brain more easily and will not cause the feared immunological adverse events typical for antibodies (amyloid-related imaging abnormalities “ARIA” indicative of local brain edema). GAL-201’s mechanism of action makes it possible to block the formation of toxic Aβ oligomers, which should be more efficient than the removal of already formed toxic species with an antibody. In addition, the high specificity of antibodies may be disadvantageous concerning a heterogeneous target such as the various species of toxic Aβ oligomers (dimers, trimers, etc.). According to the preclinical data, GAL-201 can remove the whole spectrum of possible toxic Aβ oligomers by preventing their formation.

## 4. Materials and Methods

GAL-201 (formerly also known as MRZ-14042) is a proprietary small molecule with the following IUPAC nomenclature: (2R)-2-amino-N-(1-carbamomyl-1-methlyethyl)-3-(1H-indol-3-yl)propanamide. The molecular formula of the free base is C_15_H_20_N_4_O_2_ and the molecular weight is 288.34 g/Mol. The material was manufactured by Soneas Research Ltd. in Budapest, Hungary. The last batch released, which was used in most of the experiments described below, was a fumarate salt, which is an off-white solid with a purity of 99.4% (HPLC area, certificate of analysis # QAF-10175 v1 of 3 December 2018).

### 4.1. Aβ_1-42_ and AβpE3

Aβ_1-42_ (order number H-1368; Bachem, Bubendorf, Switzerland) and AβpE_3_ (Sigma Aldrich, Burlington, MA, USA) were suspended in 100% hexafluoroisopropanol (HFIP, Sigma Aldrich) to approximately 1 mg/400 µL HFIP and shaken at 37 °C for 1.5 h. The HFIP was removed by evaporation using a Speedvac for approximately 30 min, and when completely dry, aliquots of 100 µg Aβ_1-42_ and 100 µg AβpE3 were stored at −20 °C. The Aβ_1-42_ was dissolved in DMSO (Sigma Aldrich) to a concentration of 100 µM with the aid of an ultrasonic water bath. This solution was further diluted using Ringer solution to a concentration of 50 nM Aβ_1-42_. AβpE3 was dissolved in H_2_O.

### 4.2. Binding Assay Using Surface Plasmon Resonance (SPR)

SPR experiments allow the investigation of binding of compounds to lower concentrations of Aβ_1-42_ and offer the possibility of directly assessing the affinity of such binding. A Biacore X100 SPR instrument, equipped with two flow cells on a sensor chip, was used for real-time binding studies.

Aβ_1-42_ (American Peptide Company, Sunnyvale, CA, USA) was dissolved in hexafluoro isopropanol (HFIP) to a final concentration of 1 mg/mL. The tube was tightly sealed and incubated at room temperature for 1.5 h while shaking; 100 µg aliquots were prepared in low binding Eppendorf tubes and frozen at −80 °C for 30–60 min. After lyophilization overnight, the aliquots were stored at −20 °C until use. For the preparation of monomers, one Aβ aliquot was thawed and freshly dissolved in DMSO (anhydrous), This 5 mM stock solution was centrifuged (5 min 13,000× *g*) and the supernatant diluted to 100 µM in 10 mM sodium acetate pH 4.0 immediately before immobilization.

Aβ monomers were covalently coupled to the flow cells of the CM7 sensor chips (carboxymethylated dextran matrix attached to gold surface) with 0.1 pg/mm^2^ Aβ_1-42_ density on the surface matrix [52,53]. For immobilization of human Aβ_1-42_ monomers, HFIP-treated peptide was dissolved in DMSO to 5 mM, diluted to 100 µM in 10 mM sodium acetate pH 4.0, and immediately coupled to the surface of one flow cell of the sensor chip. The second flow cell was used as a reference and treated with ethanolamine instead of Aβ. To determine affinity, GAL-201 was tested in concentrations ranging from 0.3 to 1000 nM using HBS-EP, 0.1% DMSO as a running buffer at 25 °C.

Resonance units (RUs) elicited by the compound injected into the ethanolamine control flow cell were set as reference response and subtracted from the RUs elicited by the same compound injected to the Aβ saturated flow cell. The relationships between each RU obtained at the steady state of binding (plateau of the binding curve) and each concentration of the compound were plotted. Biacore ×100 control software Ver. 1.1 (GE Life sciences, Uppsala, Sweden) was used to record the binding curves and Biacore ×100 evaluation software Ver. 1.1 to analyze them (plot each RU at the steady state vs. concentration of analyte, fit the plot, and determine K_D_ values). The dissociation equilibrium constant KD of the analyte to the immobilized Aβ was determined from the steady-state levels by estimating the maximum RU R_max_ and calculating the K_D_ as the concentration of the compound that elicited one-half of the R_max_.

### 4.3. Pharmacokinetics in Rats

The experiments resulting in Figure 2A were conducted at Brains On-Line B.V. in 2011 (Merz Pharmaceuticals internal report MD_20101011_GRA_1 as of 18 April 2011). Adult male Sprague Dawley rats (Harlan, The Netherlands) were used with a body weight 280–350 g. GAL-201 was administered orally at a dose of 50 mg/kg through gavage.

Animals were anesthetized using isoflurane 2% and oxygen, fynadine (1 mg/kg s.c.), and a mixture of bupivacaine and adrenaline for local anesthesia. The tip of the microdialysis probe (MetaQuant) was placed in the prefrontal cortex at the following coordinates: AP = 3.4 mm (from bregma), lateral = 0.8 mm (from midline), and ventral = −5.0 mm (from dura), according to the stereotactic rat brain atlas of Paxinos and Watson (1982). Blood samples were collected through a catheter placed into the jugular vein. Tubes were positioned in a way that the animals could move freely. Experiments were started 24 h after surgery. Each animal received two treatments with an interval of 3 days.

Microdialysis was initiated 1 h prior to application of GAL-201 (t = 0 min) to allow the system to stabilize. Flow rate was 0.10 μL/min with a carrier flow rate of 0.8 μL/min with a physiological perfusate fluid containing 0.2% BSA. The first 10 dialysate samples were collected in 20 min intervals. Blood samples were collected with automated equipment (AccuSamplers, DiLab, Verutech AB, Lund, Sweden) using saline with 20 IE/mL heparin as rinsing fluid.

For bioanalytics, samples were injected after preparation into an automated HPLC system using a Zorbax Eclipse (C8, 4.6 × 150 mm, 5 µm) column at a flow of 0.8 mL/min and an acetonitril gradient. MS analyses were performed on an API 4000 MS/MS detector and a Turbo Ion Spray interface (5 kV, Applied Biosystems, Bleiswijk, The Netherlands).

### 4.4. Electrophysiology

Single-cell recording. Hippocampal neurons were isolated from P5–P6 mouse brain (Neonatal mice C57BL/6N, strain code 027, Charles River) using the standardized procedure [54]. Isolated neurons were maintained for 7 days in neurobasal plating medium (containing B27 supplement, 0.5 mM glutamine, 25 μM glutamate, 10,000 units/mL penicillin, 10,000 μg/mL streptomycin, 1 mM HEPES, 10% heat-inactivated horse serum) and then the cells were added to plates. The electrophysiological recordings were performed on single cells using the perforated whole-cell configuration in current clamp mode. Data were collected using an Axopatch 200B amplifier (Molecular Device, San Jose, CA, USA) and ionic currents were digitized at 5 kHz and filtered at 1 kHz with a Digidata 1322 acquisition system. Clampex 9 was used as the acquisition software. Patch pipettes (GB150F-8P with filament, Science Products) were pulled from hard borosilicate glass on a Brown-Flaming P-87 puller (Sutter Instruments, Novato, CA, USA) and fire-polished to a final electrical resistance of 4–5 M Ω. Just before the experiments, the culture medium was substituted with the external recording solution. The perforated whole-cell configuration was achieved by using the antibiotic Gramicidin (Sigma Aldrich) diluted in the internal solution at a final concentration of 5 and 10 μg/mL for RGCs and RPEs, respectively. Electrical access to the cell was thereby achieved after about 5–10 min.

The solutions used were (in mM): external solution—140 NaCl, 2.5 KCl, 1.8 CaCl_2_, 0.5 MgCl_2_, 10 glucose, 10 HEPES, pH 7.4; internal solution—140 KCl, 10 NaCl, 2 MgCl_2_, 0.1 CaCl_2_, 10 glucose, 10 HEPES, pH 6.9.

### 4.5. In Vitro Extracellular Recording

Brain slice preparation. Transverse hippocampal slices (350 µm thick) were obtained from adult (2 months) mice that were isoflurane-anesthetized and decapitated. The experimental protocols were approved by the Ethical Committee on Animal Care and Use, Government of Bavaria, Germany. The brain was rapidly removed, and slices were prepared in ice-cold Ringer solution using a HM 650v Vibroslicer (Microm, Thermo Fisher Scientific, Waltham, MA, USA). All slices were incubated for 30 min at 37 °C and then for at least 60 min at room temperature. Slices were then transferred to a superfusing chamber for extracellular or whole-cell recordings. The flow rate of the solution through the chamber was 4 mL/min. The composition of the solution was 125 mM NaCl, 2.5 mM KCl, 25 mM NaHCO_3_, 2 mM CaCl_2_, 1 mM MgSO_4_, 25 mM D-glucose, and 1.25 mM NaH_2_PO_4_, bubbled with a 95% O_2_/5% CO_2_ mixture, and had a final pH of 7.3. All experiments were performed at room temperature. Aβ_1-42_ or AβpE3 stock solution in DMSO was added to the bath solution to give a final concentration of 50 nM (except for the dilution series, see above).

Serial dilution with Aβ_1-42_/AβpE3 and GAL-201. To test the prion-like seeding hypothesis in hippocampal slices in vitro, the serial dilution was carried out by starting with GAL-201 (500 nM) in a 10:1 stoichiometric excess over Aβ_1-42_ or AβpE3 (see Figure 3). After incubation for 20 min, the Aβ_1-42_/GAL-201 or the AβpE3/GAL-201 mixture was transferred to a freshly prepared solution with 50 nM Aβ_1-42_ or AβpE3. This dilution step was repeated 5 times, finally resulting in a concentration of 0.1 nM GAL-201. The final solution was then tested for its ability to reverse the toxic effects of Aβ_1-42_ or AβpE_3_ on LTP in hippocampal slices.

fEPSP recordings. Extracellular recordings of field excitatory postsynaptic potentials (fEPSPs) were obtained from the dendritic region of the CA1 region of the hippocampus using glass micropipettes (1–2 M Ω) filled with superfusion solution. For all recordings, both stimulating electrodes were used to utilize the input specificity of LTP and thereby allow the measurement of an internal control within the same slice. Steady baseline recordings were made for at least 60 min before application of tetanic stimuli. For LTP induction, high-frequency stimulation conditioning pulses (100 Hz; 4–5 V) were applied to the Schaffer collateral–commissural pathway via two independent inputs. Before inducing LTP, Aβ_1-42_ or AβpE3 had been applied for 90 min.

For testing prevention of toxicity, HFS was delivered from one of the electrodes under conditions in the presence of GAL-201 (at the same concentration as with the GAL-201/Aβ mixture) and potentiation of the responses was monitored for at least 60 min after the tetanus. Either Aβ_1-42_ or AβpE3 was then applied via the bath solution for 90 min before attempting to induce LTP in the second input following HFS delivered via the second electrode.

For testing detoxification, Aβ_1-42_ (50 nM) was applied via the bath solution for 90 min before attempting to induce LTP following HFS delivered via the first electrode. After recording LTP for 60 min, the bath solution was exchanged for that following serial dilution. This solution still contained 50 nM of Aβ_1-42_, but only 0.1 nM of GAL-201. This was incubated for a further 90 min before attempting to induce LTP in the second input, which was then recorded for an additional 60 min.

Amplified fEPSPs were filtered (3 kHz), digitized (15 kHz), and measured and plotted online, using the “LTP-program” software [55]; available from https://www.winltp.com, accessed on 1 March 2008). Measurements of the slope of the fEPSP were taken between 20% and 80% of the peak amplitude. Slopes of fEPSPs were normalized with respect to the 20 min control period before tetanic stimulation.

### 4.6. In Vivo LTP in the CA1 Hippocampal Region of the Anaesthetized Rat

Adult male Sprague-Dawley rats (250–400 g) were anaesthetized initially with isoflurane (5% in oxygen) and subsequently with an intraperitoneal injection of urethane (1.25 mL/100 g, 12% solution), supplemented as necessary (0.1 mL/100 g) on the basis of corneal reflex, withdrawal response to paw-pinch, and the stability of monitored cardiovascular variables. Core body temperature was monitored and maintained at 37 ± 1 °C by a homeothermic blanket system (Harvard Equipment). The right femoral vein and artery and the trachea were cannulated to permit, respectively, administration of supplemental anesthetic; the recording of arterial blood pressure via a pressure transducer and amplifier (Neurolog NL108, Digitimer, Welwyn Garden City, UK); and the maintenance of a clear airway. Animals were allowed to breathe room air, which in some cases was oxygen-enriched. Animals were placed in a stereotaxic frame (Narishige ST-7) and the dorsal brain surface overlying the hippocampus was exposed by craniotomy. Small incisions were made in the dura and stimulating and recording electrodes were lowered vertically through the cortex to the hippocampus according to the following stereotaxic coordinates: recording electrode; bregma −4.4 mm, lateral 2.0–2.25 mm, depth 2.0–2.7 mm below pial surface; stimulating electrode; bregma −3.4 mm, lateral 2.5 mm, depth 2.0–3.0 mm below pial surface. Finally, an i.c.v. cannula (stainless steel, gauge 30; Semat Inc., Taranto, Italy) was lowered according to the following stereotaxic coordinates; bregma +0.5, lateral 1.5, 3.7 mm below pial surface with a 17° rostro-caudal angle for injection into the lateral ventricle [56].

Electrical stimulation (0.1–0.2 ms pulse width, 10–100 V, 0.1 Hz) of the Schaffer collateral pathway though a coaxial bipolar stainless-steel electrode (FHC) was used to evoke population spike activity in stratum pyramidale of area CA1, recorded through an extracellular carbon fiber microelectrode (Kation Scientific). Using Neurolog equipment, signals were amplified (×0.2–0.5 k, NL102, NL104) and filtered (bandwidth 10 Hz −1 kHz, NL125) with the conditioned output being captured on a PC using a micro1401 interface with Spike 2 software (Cambridge Electronic Design, Cambridge, UK). The amplitude of the population spike (PS) superimposed on the field excitatory postsynaptic potential was presented in real time and subsequently analyzed off-line. By adjusting the depth of both the stimulating and recording electrodes in small increments, the amplitude of the PS was optimized. Thereafter, an input–output curve was conducted to determine maximal PS amplitude and the voltage required to generate a potential with amplitude of approximately 30–50% of the maximum. Stimulation parameters were then maintained at this level at a frequency of 0.033 Hz to demonstrate a stable baseline period of at least 10 min before commencing the full experiment protocol (Figure 9). In each experiment, all data points were normalized to the mean of the PS amplitude over the 10 min period directly before TBS. For statistical purposes, this baseline period was compared to the last 10 min of recording (80–90 min post-TBS). Potentials were recorded for 90 min post-TBS.

### 4.7. Preparation of Test Compound and Oligomeric Aβ_1-42_

GAL-201 and Aβ_1-42_ were prepared on the day of use. GAL-201 was dissolved in PBS and immediately loaded into a 1 mL syringe topped with a 25-gauge needle tip, inserted under the skin overlaying the back and injected in a 2 mL/kg volume to give a 0.08–10 mg/kg GAL-201 dose. Aβ_1-42_, supplied in 5 µg aliquots and stored at −20 °C, was warmed in a water bath at 37 °C for 10 min, and then sonicated for 30 s, diluted in 667 µL PBS to start the oligomerization process and sonicated for a further 30 s, vortexed for 30 s, sonicated for 30 s, and vortexed one final time for 30 s before being placed on ice. Oligomeric Aβ_1-42_ solution was used between 15 and 45 min following preparation and was brought to room temperature before use by loading into the i.c.v. cannula 10 min before administration. One 6 µL pump cycle was performed prior to cranial insertion of the i.c.v. cannula, to ensure immediate flow.

### 4.8. Administration of Test Compound and Oligomeric Aβ_1-42_

Ten minutes subsequent to the start of recording, s.c. injection of GAL-201 (0.08–10 mg/kg) or vehicle (PBS: H_2_O; 4:1, 2 mL/kg) was performed. Thirty minutes following s.c. compound/vehicle administration, an i.c.v. injection of 6 µL of oligomeric Aβ_1-42_ or PBS was administered over a 3 min period. Following i.c.v. injection, the signal was recorded for a further 120 min before 5× theta burst stimulation (TBS, Figure 1) was applied to the Schaffer collateral pathway.

### 4.9. Statistical Analysis

All data in this study are presented as mean ± SD. In each experiment, all data points were normalized to the mean of the PS amplitude over the 10 min period directly before TBS. For statistical purposes, this baseline period was compared to the last ten minutes of recording (80–90 min post-TBS). The probabilities of significant difference between groups were calculated in GraphPad Prism 5 software, using one-way ANOVA (one-way analysis of variance) with a post hoc Bonferroni’s multiple comparison test and using Student’s *t*-test for probability (*p*) values displayed in Table 1; *p* values of <0.05 were considered to represent significant differences.

## 5. Conclusions

In comparisons with the other developmental AD treatment approaches targeting soluble Aβ oligomers, GAL-201 combines several favorable attributes for AD therapy. These include the practical advantages of a small molecule, sufficient central availability with oral administration, high affinity for the target Aβ_1-42_, detoxification also of AβpE3, exceptionally low stoichiometric requirements (five-fold stoichiometric undersupply effective), immediate functional improvement of synaptic plasticity (LTP), reversal of already established neurotoxic Aβ_1-42_ effects, and the “trigger effect” enabling long-term therapeutic security even at low intake compliance.

With the mechanism of action described in this manuscript, two types of clinical effects can be expected in AD patients undergoing oral GAL-201 treatment. First, a short-term effect based on immediate improvement of synaptic plasticity, which translates into rapid cognitive improvement. Second, a long-term effect based on reduced Aβ_1-42_ neurodegeneration, reflected in a slowdown or even a halt of AD progression. In summary, the data available to date warrant the initiation of IND-enabling studies to advance the development of GAL-101 towards clinical studies.

## Figures and Tables

**Figure 1 ijms-23-05794-f001:**
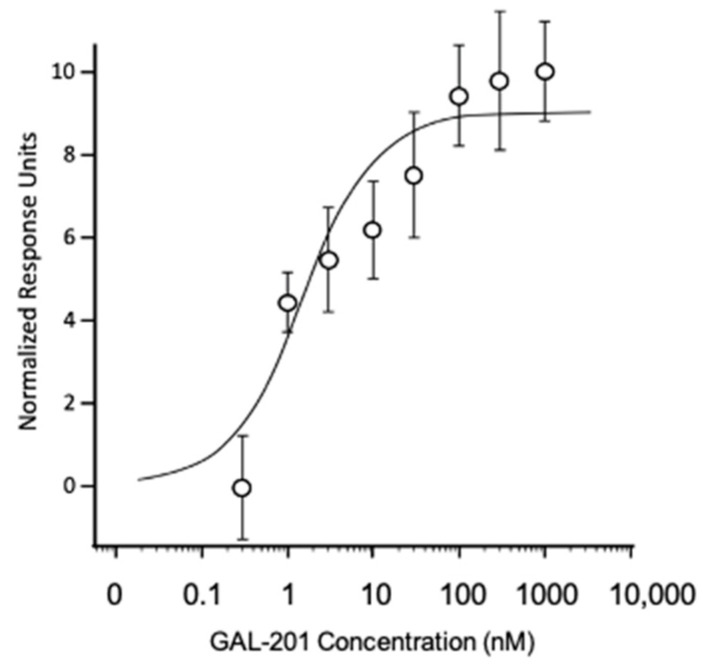
GAL-201 binding to Aβ_1-42_ using surface plasmon resonance (SPR). This binding assay was performed using a Biacore X100 biosensor instrument equipped with two flow cells. Aβ_1-42_ was covalently coupled to one flow cell of CM7 sensor chips using an amine coupling kit. HBS-EP was used as assay running buffer. The analyte GAL-201 was injected over the sensor chip in concentrations ranging from 0.3 to 1000 nM at a flow rate of 10 µL/min for 180 s at 25 °C. Responses were evaluated at steady state, plotted against concentration, and fit by the four-parameter logistic equation to a K_D_ = 2.5 nM (n = 4). χ^2^ = 6.124.

**Figure 2 ijms-23-05794-f002:**
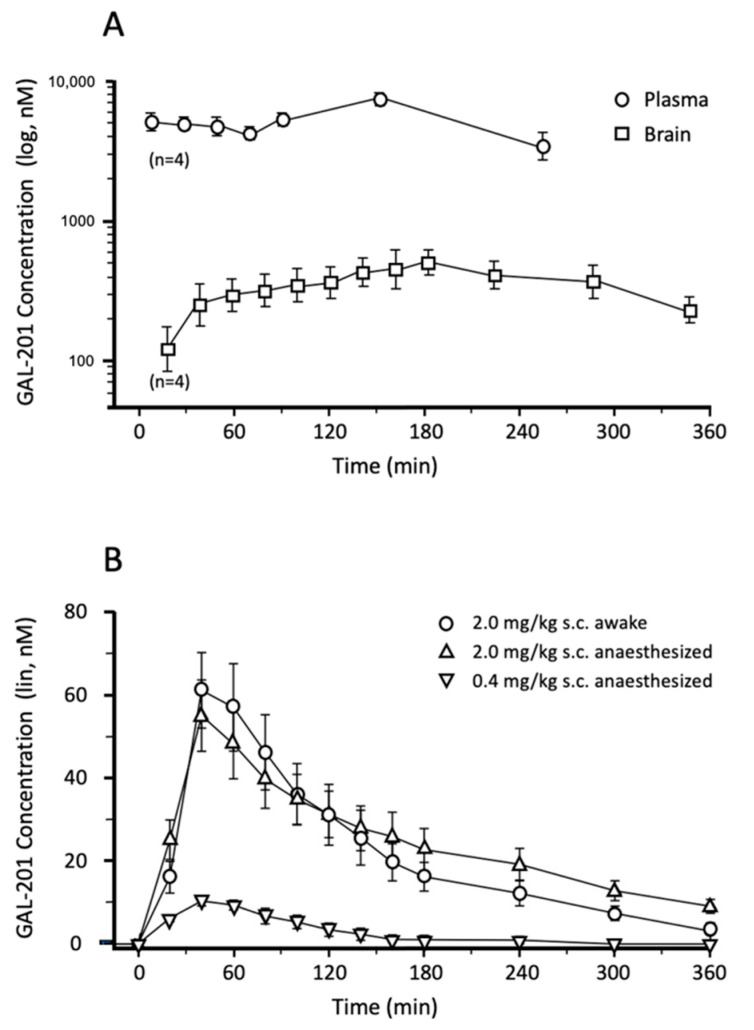
Pharmacokinetics of GAL-201 in rats. (**A**) 50 mg/kg GAL-201 was administered orally to adult male Sprague Dawley rats. A microdialysis probe was placed in the prefrontal cortex to measure the concentration of GAL-201 in the brain interstitial fluid (ISF). Blood samples were collected through a jugular catheter with automated equipment. The animals were able to move freely. Samples were analyzed using HPLC with MS/MS detector. GAL-201 concentrations in ISF dialysate (Brain, squares) and in blood plasma (Plasma, circles) are depicted on a logarithmic scale as means of 4 measurements ±95% confidence limits. (**B**) This experiment was conducted under similar conditions as described in panel (**A**). However, GAL-201 was administered as subcutaneous (s.c.) injection at doses of 0.4 mg/kg and 2.0 mg/kg. The animals were either anesthetized or awake and freely moving. Microdialysis data are depicted on a linear scale as means of 4 measurements ±SEM).

**Figure 3 ijms-23-05794-f003:**
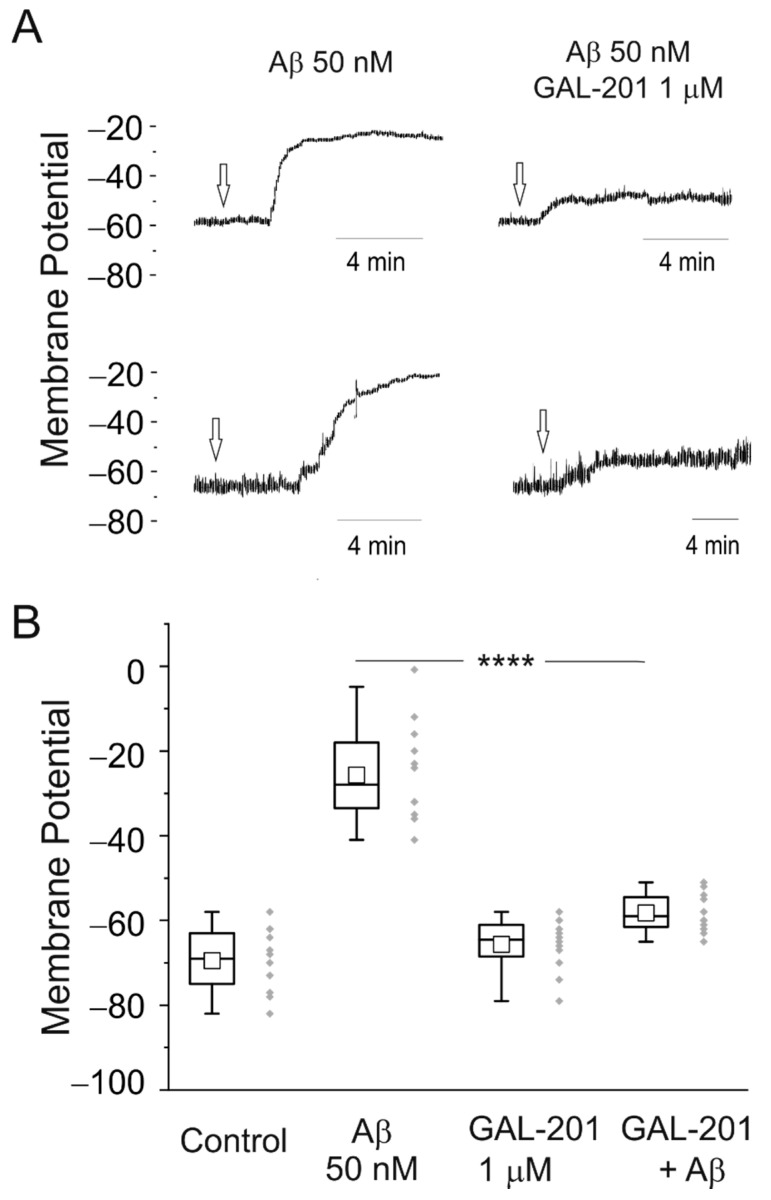
GAL-201 prevents the Aβ_1-42_−induced depolarization in isolated mouse neonatal hippocampal neurons. (**A**) Examples of time course recordings of membrane resting potential at 23 °C. After monitoring the membrane voltage for a few minutes in control condition, Aβ_1-42_ was perfused (arrows) inducing a strong depolarization (left) while in the presence of GAL-201 (right) depolarization reaches only a few millivolts. (**B**) Average resting potential measurements in control, perfusing 50 nM Aβ_1-42_ plus 1 µM GAL-201 (n = 12 each condition; **** *p* < 0.0001).

**Figure 4 ijms-23-05794-f004:**
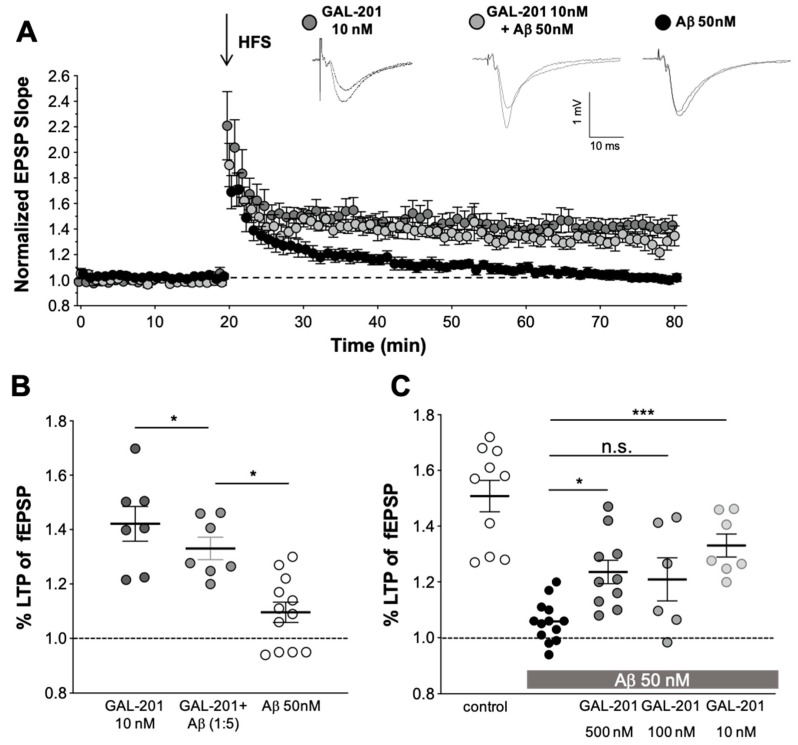
GAL-201 prevented the Aβ_1-42_-induced LTP blockade in vitro at low concentrations. (**A**) Slices were superfused at 23 °C with 10 nM GAL-201 solution for 90 min before HFS was delivered in the first input (dark gray circles). At that concentration, GAL-201 allowed the induction of LTP. To ensure the validity of the Aβ_1-42_-mediated effect, interleaved experiments with Aβ_1-42_ (50 nM, closed circles, n = 12) were additionally performed in separate slices. In the presence of Aβ_1-42_, GAL-201 at 10 nM was able to prevent LTP blockade (light gray circles). Representative fEPSPs are shown on top. (**B**) The magnitude of LTP potentiation in the presence of 10 nM GAL-201, 50 nM Aβ_1-42_ alone, and 10 nM GAL-201 together with 50 nM Aβ_1-42_ is shown as a scatter dot plot representing the potentiation of the fEPSP slope values averaged from the last 10 min of the recordings of each single experiment. The effect of 10 nM GAL-201 in the presence of 50 nM Aβ_1-42_ was significant to 50 nM Aβ_1-42_ alone. (**C**) Scatter dot plot summarizing the last 50 to 60 min after HFS. Effects of different stoichiometric Aβ_1-42_/GAL-201 ratios (10:1, 2:1, 1:5) on their ability to prevent LTP reduction were investigated. HFS potentiated fEPSP under control conditions and Aβ_1-42_ almost blocked LTP. GAL-201 (500 and 10 nM) significantly prevented LTP in the presence of Aβ_1-42_. When 100 nM GAL-201 was applied, LTP showed a higher potentiation as with 50 nM Aβ_1-42_; however, the effect was not significant. (* *p* < 0.05, *** *p* < 0.001).

**Figure 5 ijms-23-05794-f005:**
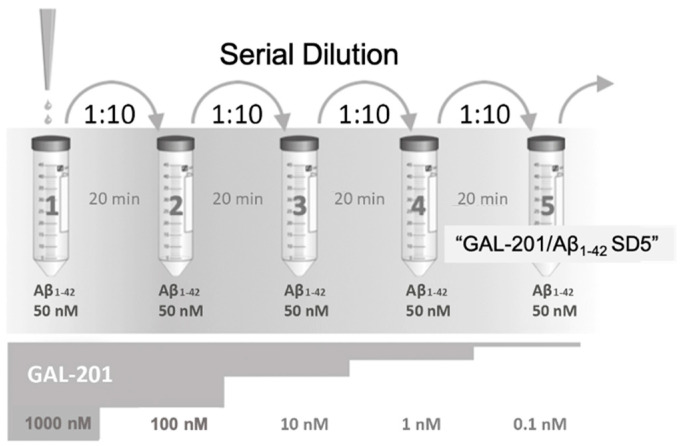
Serial dilution of GAL-201 in the presence of Aβ_1-42_. Each of the vials #1 through #5 initially contained toxic Aβ_1-42_ oligomers formed from 50 nM Aβ_1-42_. GAL-201 was pipetted into vial #1 only, resulting in a starting concentration of 1000 nM GAL-201. After 20 min, an aliquot was removed from vial #1 and transferred to vial #2, diluting GAL-201 by 1:10. After another 20 min this dilution step was repeated again, finally leading to a GAL-201 concentration of 0.1 nM in vial #5 (dilution 1:10,000). The concept behind this “serial dilution” is to gradually remove the GAL-201 almost completely from the solution so that the remaining concentration in the last vial is far below its target affinity and can no longer exert any direct pharmacological activity itself. For the solution in the last vial #5 the term “GAL-201/Aβ_1-42_ SD5” is used throughout this manuscript. Interestingly, after this procedure the solution in vial #5 was no longer toxic in LTP experiments. Since this detoxification cannot be attributed to the remaining GAL-201 traces (0.1 nM), it is obviously due to other components of the solution, presumably Aβ_1-42_ species that were initially formed in the presence of higher concentrations of GAL-201. It is postulated that these Aβ_1-42_ species act as seeds for the further agglomeration of misfolded Aβ_1-42_, which leads to a self-propagating detoxification (“trigger effect”). The same procedure was also applied using AβpE3 instead of Aβ_1-42_ with identical results.

**Figure 6 ijms-23-05794-f006:**
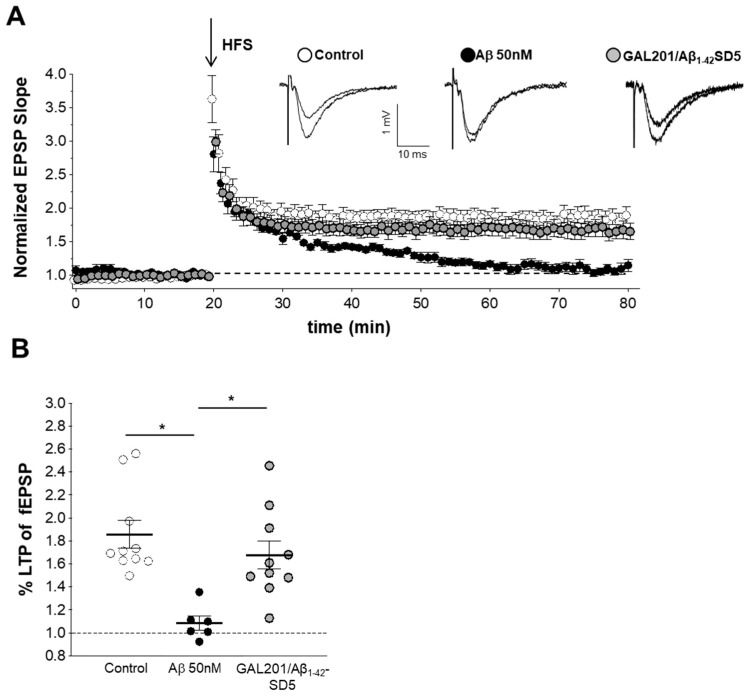
GAL-201 prevented synaptotoxic effects of Aβ_1-42_ on LTP after serial dilution. (**A**) Normalized fEPSP time course following a HFS under control conditions, with 1.5 h Aβ_1-42_ exposure alone and the concomitant application of GAL-201 and Aβ_1-42_ via a GAL-201/Aβ_1-42_-SD5 solution. The insets show representative fEPSP traces of a control and Aβ_1-42_ measurement. (**B**) Scatter dot plot summarizing the last 50 to 60 min after HFS. The fEPSP amplitude was significantly reduced by Aβ_1-42_ (50 nM). After the “serial dilution” of GAL-201 by factor 1:10,000 (see Figure 3), the GAL-201/Aβ_1-42_-SD5 solution from vial #5 reversed Aβ_1-42_ synaptotoxicity. This is explained by the formation of self-propagation Aβ_1-42_ species since the trace concentration of only 0.1 nM GAL-201 itself has been proven not to be directly pharmacologically active [34]. (* *p* < 0.05).

**Figure 7 ijms-23-05794-f007:**
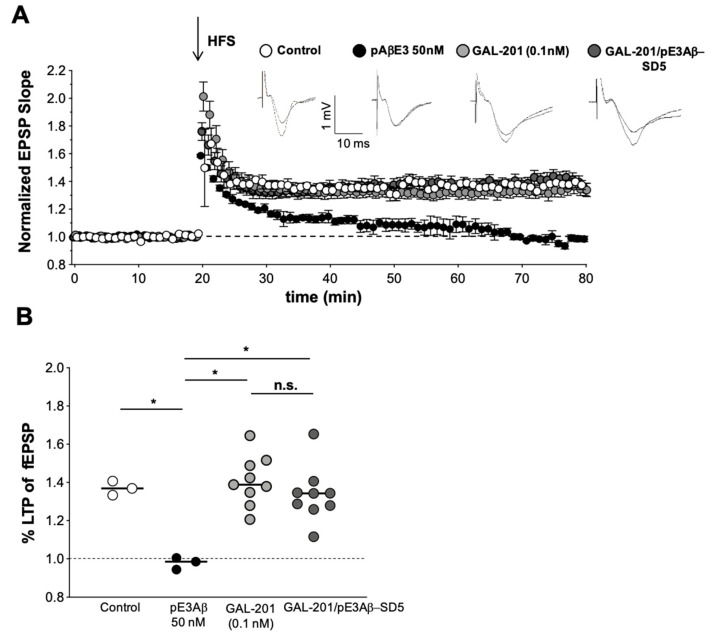
The toxic effects of AβpE3 (50 nM) upon LTP induction can be prevented when using GAL-201/AβpE3-SD5 solution from serial dilution. (**A**) Under control conditions, LTP was induced in one input (open circles) and in the presence of 50 nM of AβpE3 (closed circles). These interleaved experiments have been performed to ensure the validity of the AβpE3-mediated effect. The strong synaptotoxic effect of AβpE3 was demonstrated as LTP was completely blocked. In a separate slice, LTP has been delivered (input one) in the presence of 0.1 nM GAL-201 (light grey circles) which did itself not affect LTP. The GAL-201/AβpE3-SD5 solution generated from AβpE3 (initiated with GAL-201) through serial dilution was applied 90 min before HFS delivery in the second input (dark gray circles) and prevented LTP blockage. Representative fEPSPs are shown on top. (**B**) Scatter dot plot summarizing the last 50 to 60 min after HFS for all groups. (* *p* < 0.05).

**Figure 8 ijms-23-05794-f008:**
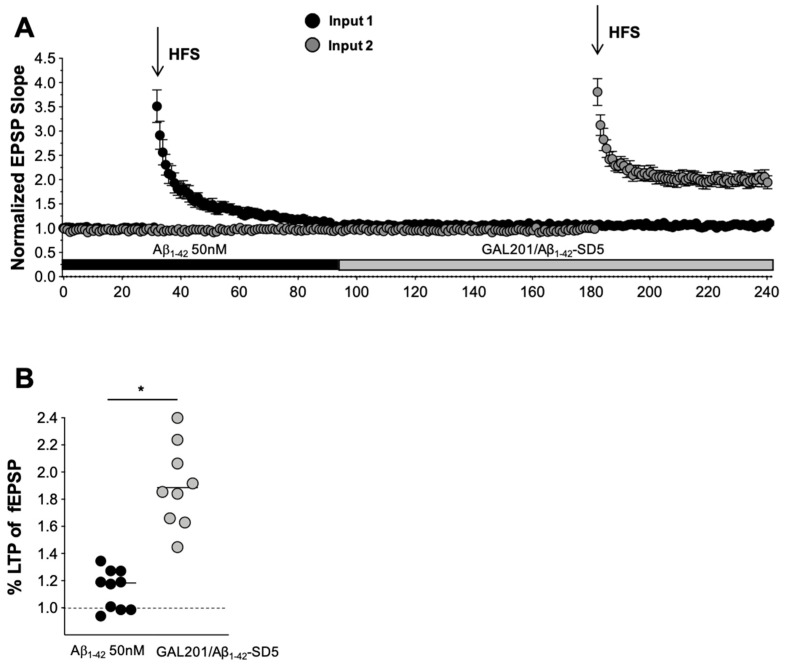
Reversal of an already established synaptotoxic effect caused by Aβ_1-42_. The GAL-201/Aβ_1-42_-SD5 solution was capable of detoxifying an already established neurotoxic process. (**A**,**B**) LTP was first induced in the presence of 50 nM Aβ_1-42_ in the first input then after the application of the GAL-201/Aβ_1-42_-SD5 solution in the second input. (**A**) Pooled data of the full-length two-input experiments. After slices were incubated with GAL-201/Aβ_1-42_-SD5 solution, LTP completely recovered. (**B**) Scatter dot plot summarizing the experiments for GAL-201 when the last 50 to 60 min after HFS were analyzed. (* *p* < 0.05).

**Figure 9 ijms-23-05794-f009:**
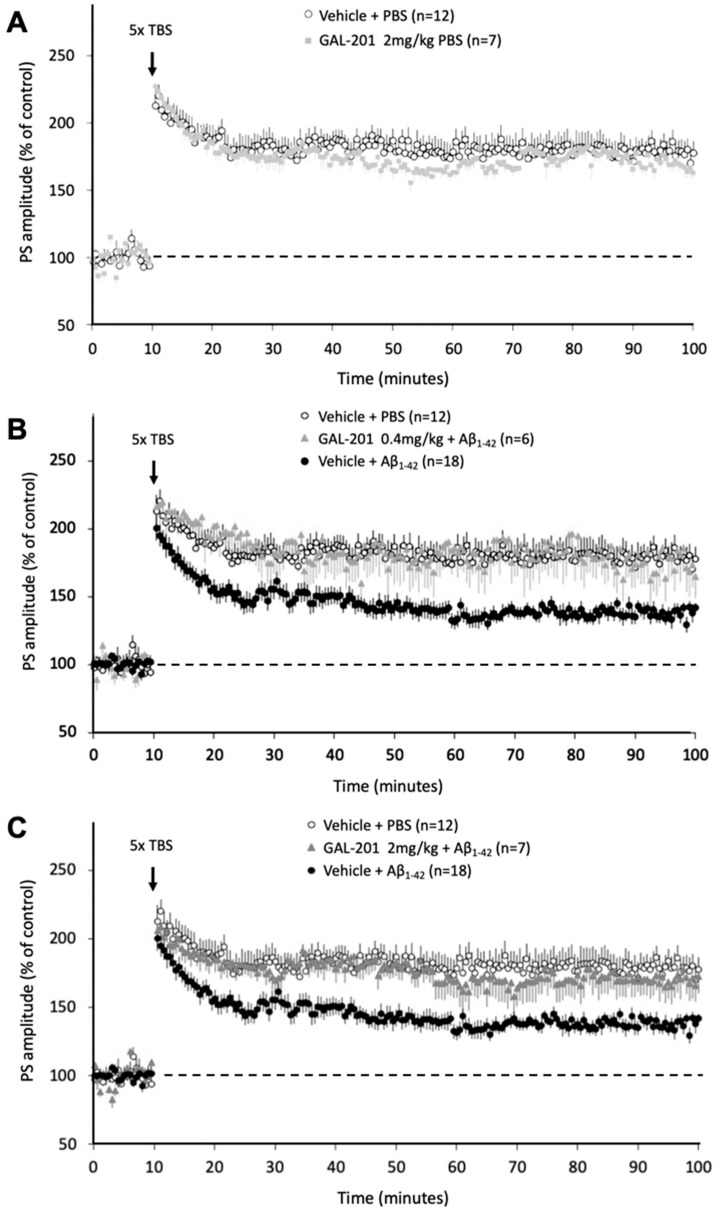
In vivo LTP in rats with i.c.v. applied Aβ_1-42_ and subcutaneously applied GAL-201. (**A**) Preliminary studies tested the effects of a 2 mg/kg dose of s.c. administered GAL-201 on TBS-induced hippocampal LTP, in the absence of oligomeric Aβ_1-42_. GAL-201 2 mg/kg itself did not confer any significant inhibitory or facilitatory effect on hippocampal synaptic responses following theta burst stimulation. (**B**) In animals subjected to i.c.v. injection of 6 µL oligomeric Aβ_1-42_ preceded by s.c. vehicle, significant deficits in LTP were observed. In animals subjected to *i.c.v* injection of 6 µL oligomeric Aβ_1-42_, a preadministration of s.c. GAL-201 2 mg/kg protected against oligomeric Aβ_1-42_-induced deficits in hippocampal LTP. (**C**) In animals subjected to *i.c.v* injection of 6 µL oligomeric Aβ_1-42_, a preadministration of s.c. GAL-201 0.4 mg/kg demonstrated a protection against oligomeric Aβ_1-42_-induced deficits in hippocampal LT.

**Figure 10 ijms-23-05794-f010:**
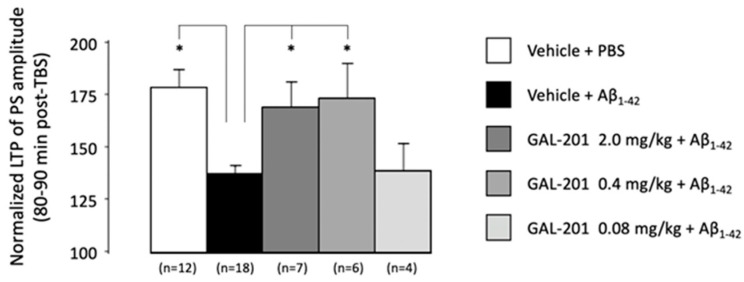
In vivo LTP in rats with i.c.v. applied Aβ_1-42_ and subcutaneously applied GAL-201. Bar chart detailing the LTP of PS amplitude 80–90 min after theta burst stimulation for the various treatment groups. Data are expressed as a percentage of the pre-TBS control ±SEM. In animals subject to *i.c.v* injection of 6 µL oligomeric Aβ_1-42_, a preadministration of s.c. GAL-201 0.08–2 mg/kg produced a dose-dependent protection against oligomeric Aβ_1-42_-induced deficits in hippocampal LTP, the highest GAL-201 doses (0.4 and 2 mg/kg) bringing post-TBS responses towards those observed in animals receiving i.c.v. PBS. (* *p* < 0.05).

**Table 1 ijms-23-05794-t001:** Reversal of stoichiometric ratio potentiates detoxifying strength.

Concentration GAL-201	500 nM	100 nM	10 nM
Stoichiometric ratio GAL-201:A_1-42_	10:1	2:1	1:5
Relative effect size (blocking A_1-42_ toxicity)	40.0%	33.3%	60.0%

## Data Availability

Not applicable.

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
