# Peer review of "The Small Molecule GAL-201 Efficiently Detoxifies Soluble Amyloid β Oligomers: New Approach towards Oral Disease-Modifying Treatment of Alzheimer’s Disease"

_ijms, 2022, doi:10.3390/ijms23105794_

Round 1

Reviewer 1 Report

The manuscript by Russ et al. shows how GAL-201 (a small molecular weight compound) can modulate the toxicity of Aβ­­­42­ ­­oligomers. This is especially important because the number of those who suffer from AD increases every year. This study includes the preparation of Aβ­­42 and AβpE3 oligomers, pharmacokinetics in rats and other interesting experiments. However, before supporting this publication, comments have to be addressed to help the authors improve their manuscript.

  1. Could the authors clarify the phrase mentioned in the abstract - “misfolded Aβ1-­­42 monomers” (line 25)? Note: Amyloid-beta belongs to the class of intrinsically disordered proteins with no stable secondary structure, meaning that it is metastable in aqueous solutions. Did the authors mean to write misfolded oligomers?
  2. The discussion of the article includes the literature overview that should be placed in the introduction section. For example, the first two paragraphs of the discussion do not consider any aspect of the results. However, the text is well written and informative. This text should be written down in the introduction of the article.
  3. In the figures 4B, 4C, 6B and 7B, the authors present the scatter dot plots, but the data array has an uneven number of points. This leads to shifted evaluation. For example, in figure 4B, the Aβ 50nM has four dot points (4 out of 12 presented) with a value similar to the four points of GAL-201+Aβ (1:5) (4 out of 7 presented). With such a small array of data points, it may seem that the experiment results may be accidental. Could the authors explain why they did not take an equal number of data points?

Minor:

  1. β sign is missing in line 31.
  2. Did the authors mean 1 µM GAL-201 (instead of 1 mM GAL-201) in Figure 3 B?
  3. In line 198, the brackets are missing a symbol.
  4. The authors should indicate the coefficient of determination value of their sigmoidal fit.

Author Response

Referee 1:

Comment by referee:

Could the authors clarify the phrase mentioned in the abstract – “misfolded Aβ1-­­42 monomers” (line 25)? Note: Amyloid-beta belongs to the class of intrinsically disordered proteins with no stable secondary structure, meaning that it is metastable in aqueous solutions. Did the authors mean to write misfolded oligomers?

Reply by the authors:

Our understanding is based on the work of Frydman-Marom et al. (Angew Chem Int Ed Engl 2009, 48, (11), 1981-6) which has been further substantiated by our experiments. The ability of amyloid beta to aggregate is depending on a specific conformation (secondary structure) resulting in the known beta-sheet aggregates. Several authors call this “misfolded amyloid beta” with occurs also as a transient monomer, before self-stabilization through the aggregation. GAL-201 has been designed to bind specifically to this misfolded form (see above-cited paper), which is definitely “metastable” as mentioned by the referee. This conformational change affects the monomeric molecules as they change from a-helix to b-sheet. The isoform Ab1-42 is very much prone for this change as it shows in its toxicity.

Comment by referee:

The discussion of the article includes the literature overview that should be placed in the introduction section. For example, the first two paragraphs of the discussion do not consider any aspect of the results. However, the text is well written and informative. This text should be written down in the introduction of the article.

Reply by the authors:

The authors appreciate this helpful comment and have restructured the introduction and discussion section accordingly. The introduction is now longer and contains the literature overview from the first sections of the previous discussion. The new discussion section is now focusing more on the interpretation of the data. This restructuring also enabled a reduction of redundancy.

Comment by referee:

In the figures 4B, 4C, 6B and 7B, the authors present the scatter dot plots, but the data array has an uneven number of points. This leads to shifted evaluation. For example, in figure 4B, the Aβ 50nM has four dot points (4 out of 12 presented) with a value similar to the four points of GAL-201+Aβ (1:5) (4 out of 7 presented). With such a small array of data points, it may seem that the experiment results may be accidental. Could the authors explain why they did not take an equal number of data points?

Reply by the authors:

We thank the referee for raising this point. As outlined in the Method section, for LTP recordings we were using the hippocampal slice preparation where the neurons are distributed in a laminar order. This neuroanatomical morphology provides the advantage to induce LTP via two independent inputs. That means we can use a control LTP and an “effect” LTP, in the presence of a changed condition e.g., the application of a drug; but only two stimulations are feasible.

In the mentioned figures (4, 6 and 7) we plotted more than two data sets, e.g., for Fig. 4 the results from 3 stimulations. According to the explanation above, the groups “GAL-201 10 nM and “GAL-201+Aβ (1:5)” has been performed in one slice. The “Aβ (50 nM)” group has been conducted in interleaved experiments in different slices where we induced a control LTP (not shown) and LTP in the presence of Aβ (50 nM). Due to this design, the presented number of points has an uneven number. This holds for Fig. 4 and 6. In Fig. 7 we present scatter dot plots pairwise (Control/ pE3Aβ 50 nM and GAL-201 0.1 nM/GAL-201/pE3Aβ-SD5) and you see that the pairs are showing even numbers.

Comment by referee:

Minor:

β sign is missing in line 31.  Corrected

Did the authors mean 1 µM GAL-201 (instead of 1 mM GAL-201) in Figure 3 B? Yes, corrected

In line 198, the brackets are missing a symbol. Corrected. The symbol is the Greek letter omega for the electrical resistance “Ohm”: 1–2 MW

The authors should indicate the coefficient of determination value of their sigmoidal fit.

Actually, a coefficient of determination value only applies in our opinion when doing linear regression fits. The analysis program IGOR provided the chi-square value which is 6.124.

Reply by the authors: See above in blue

Reviewer 2 Report

The current manuscript is a well organized research article showing new approach towards oral disease modifying treatment of Alzheimer’s Disease using GAL-201. The manuscript needs to be improved in some ways.

Abstract should be rewritten. It should clearly mention the background, methods, results, conclusion briefly.

Introduction should mention the background, research question or hypothesis at the end. Here the authors mentioned that they presented data on detoxification effects of GAL-201 etc citing some articles, seems like previously published results. Please rewrite the section. 

In result section, most of the figures are very small, difficult to see, some texts are too small to see. Please provide larger images. Could be use colors to clearly demonstrate the results. 

Conclusion is too big, it seems to be a small discussion. Please include it at the end of discussion and briefly write a conclusion without citations. 

If possible, please include a graphical abstract or summary. 

Author Response

Referee 2:

Comment by referee:

Abstract should be rewritten. It should clearly mention the background, methods, results, conclusion briefly.

Reply by the authors:

The abstract has been modified according to the elements mentioned by the referee. Some additional high-level information about the applied methods have been added to strengthen the methods part. 

Comment by referee:

Introduction should mention the background, research question or hypothesis at the end. Here the authors mentioned that they presented data on detoxification effects of GAL-201 etc citing some articles, seems like previously published results. Please rewrite the section.

Reply by the authors:

This comment is in line with one of the comments by referee 1. The Introduction section has been re-written according to the commendations. This is the first manuscript publishing data about GAL-201.

Comment by referee:

In result section, most of the figures are very small, difficult to see, some texts are too small to see. Please provide larger images. Could be use colors to clearly demonstrate the results. 

Reply be the authors:

The authors agree with this observation by the referee. There was obviously a technical problem with some of the figures while transforming them into the IJMS format. The original figures show much better quality. Some of the figures have already been replaced (higher resolution) and magnified. It turned even out that some panel B figures got entirely lost. All that will be brought in perfect shape prior to publication.

Comment by referee:

Conclusion is too big, it seems to be a small discussion. Please include it at the end of discussion and briefly write a conclusion without citations. 

Reply by the authors:

The Conclusion has been shortened significantly as recommended by the referee, and is now without any citations. Parts of the former Conclusion section are now included into the Discussion, particularly the comparison of GAL-201 with other Aβ oligomer targeting drugs.

Comment by referee:

If possible, please include a graphical abstract or summary.

Reply by the authors:

According to the editor, graphical abstracts are not common policy in IJMS.